# Impact of Magnetic Field Environment on the EDM Performance of Al-SiC Metal Matrix Composite

**DOI:** 10.3390/mi12050469

**Published:** 2021-04-21

**Authors:** Timur Rizovich Ablyaz, Preetkanwal Singh Bains, Sarabjeet Singh Sidhu, Karim Ravilevich Muratov, Evgeny Sergeevich Shlykov

**Affiliations:** 1Mechanical Engineering Faculty, Perm National Research Polytechnic University, 614000 Perm, Russia; karimur_80@mail.ru (K.R.M.); kruspert@mail.ru (E.S.S.); 2Mechanical Engineering Department, IKG Punjab Technical University, Kapurthala 144603, India; preetbains84@gmail.com; 3Mechanical Engineering Department, Beant College of Engineering and Technology, Gurdaspur 143521, India; sarabjeetsidhu@yahoo.com

**Keywords:** electrical discharge machining, metal matrix composite, Taguchi, micro-hardness, surface roughness, material removal rate

## Abstract

In the present work, a hybrid magnetic field assisted powder mixed electrical discharge machining had been carried out on the Aluminum-Silicon Carbide (Al-SiC) metal matrix composite. The aim of the study was to obtain higher surface finish, and enhanced material removal rate. The dielectric mediums employed were plain EDM oil, SiCp mixed and graphite powder mixed EDM oil for flushing through the tube electrode. The magnetic field intensity, discharge current, T-on/off duration and type of dielectric were the control variables used for present investigation. From the results, it was observed that the machining variables for instance, discharge current, T-on/off duration and type of dielectric conditions remarkably affected the material removal rate, micro-hardness and surface roughness of the machined composite material. The MRR augmented considerably with an increase in the magnetic field intensity along with peak current. Subsequently, the composite with lesser vol.% of SiC particulates witnessed sharp rise in MRR in maximum magnetic field environment (0.66T). In addition, quality of the machined surface improved significantly in graphite powder mixed dielectric flushing condition with intermediate external magnetic field environment. Besides, an enhancement of micro-hardness was quantified as compared to base material due to the transfer of the material (SiCp) during powder mixed ED machining.

## 1. Introduction

Metal Matrix Composites (MMCs) are the lightweight materials possessing enormous exceptional mechanical and physical properties such as high specific strength, stiffness and wear resistance [1,2,3]. SiC particles reinforced aluminum matrix (Al-SiC) is one of the categories of MMCs finding huge applications in the aerospace and automotive sectors and various other industries. The machining of difficult-to-cut materials such as ceramics, super alloys and composites has been a major challenge for manufacturing industries in today’s era. New and advanced machining techniques as “non-conventional machining methods” have been the breakthrough to curb the engineering challenges posed by the rapid growth in the development of such materials. These days, these non-conventional machining processes are more commonly employed in the manufacturing for machining such hard-to-process materials class. One such machining method- EDM, is being widely used for machining all kinds of metallic materials to the desired size and shape accurately regardless of their mechanical properties [4]. It is a thermoelectric technique wherein, material removal and desired surface roughness from the workpiece is achieved by the spark generated repeatedly between tool and workpiece submerged in dielectric. Indeed, the selection of suitable machining parameters is of utmost important to achieve optimal machining responses [5,6]. For instance, as reported [7] surface roughness depended capillary imbibitions phenomenon is very important parameters in many applications of petroleum engineering, textile industries or fuel cell etc. Thus, these desired roughness can easily controlled with the suitable tuning of EDM spark energy parameters. Furthermore, aerospace related prototypes and products, automobile industry, electronics, medical and surgical components are the few prominent fields where applications of this manufacturing technology can be witnessed [8]. Additionally, desired surface modification or topology of numerous parts in medical, aeronautical and aerospace industry has been achieved using the optimized EDM process [9]. 

However, the reliability and safety of conventional ED machined components is being questioned repeatedly. The surface cracking of the machined components owing to the development of recast layer (also known as white layer) and heat affected zone underneath it has been the prominent issue. The hybrid machining (mixing of two or more conventional machining processes) has proved to be a key future technology to solve this issue. Owing to this, it’s never been an uphill task to achieve desired surface integrity as complex shapes and precision with the advent of hybrid EDM. The hybrid EDM is a mechanism to enhance the machining characteristics of a conventional EDM by combining various individual processes together to achieve desired surface integrity and stability of process and to overcome the limitations of the individual constituents [10,11]. From the research studies [12], it is witnessed that the magnetic field incorporation with conventional EDM technique significantly improves the thermoelectric properties, enhancing its machining capability. In addition, various researchers have experimented with EDM to improve for its low MRR, and enhance the surface properties of workpiece by mixing various types of powders in dielectric fluid (known as powder mixed EDM), which has proved to be a major breakthrough [13,14,15]. Likewise, an increased MRR and micro-hardness of machined surface along with reduced roughness has been witnessed in powder mixed hybrid EDM by researchers [16]. The research community has also been successful in achieving mirror-finish surfaces with a powder-mixed dielectric [17,18]. 

A novel hybrid EDM methodology using the external magnetic field has showcases its process effectiveness by enhancing the effective debris expulsion from the melt pool. Some researchers and academicians have put forward the impact of magnetic field on MRR and Surface Roughness (SR) [19] experimentally in the spark machining process. Moving a step forward, Heinz et al. [20] fully investigated the possibility to employ external magnetic field to generate a ‘Lorentz Force’ (LF) affecting the melt pool as a resultant of mutually perpendicular electric and magnetic fields. Recently, Rouniyar and Shandilya [21] experimented with an identical hybrid EDM process assisted by external magnetic field and highlighted that a resultant Force (RF) was witnessed. They demonstrated that the RF was due to the interplay of Magnetic Field (MF) and Electric Field (EF), which as a result increased the density of electrons and enhanced MRR.

This elaborative literature study encouraged the authors to dive deep into this magnetic and electric field interference to explore and uncover its process capabilities. The review of past work recognized the noteworthy effects of magnetic field on the EDM process parameters. Therefore, we aimed the present study towards the investigation of the effects of Magnetic field on the output parameters of Al-SiC composite when coupled with EDM. The quest for more efficient machining techniques necessitated the requirement of hybridization of state-of-the-art technologies with exceptional efficiency and stability. The main motive behind this study was to analyze the performance of magnetic field environment on powder mixed assisted EDM operation on machining characteristics for instance, MRR, MH, and SR of Al-SiC metal matrix composite. The prominent process parameters like peak current, duration of T-on/off, and magnetic field intensity were selected to analyze their effects on process performance.

## 2. Experimentation 

In this study, aluminum matrix-based composite (Table 1) variants reinforced with silicon carbide (SiCp, electronic grade manufactured by CPS Technologies, Norton, MA, USA) were used as workpiece. The distribution of SiC (yellow arrows) in alumnium matrix is presented in Figure 1.

The electrolytic copper tube electrode (Figure 2) having 15 mm outer and 4 mm inner diameter was used to allow the powder mixed dielectric to flow through the tool. The experimentation was carried out employing the ZNC EDM (Make: OSCARMAX, Taichung City, Taiwan) (Figure 3a). As the past studies witnessed the application of magnetic field enhanced the plasma ionization in EDM spark zone and controlled its expansion, same principle was implemented in present study. Two sets of permanent magnets (0.33 T each) were fixed in such a manner so as to achieve desired magnetic field effect into the machining zone, as shown in Figure 3b. Commercial grade EDM oil (specific gravity = 0.763, freezing point = 94 °C) was used as the dielectric medium. Figure 3c presents the arrangement of dielectric flow through the hollow electrode. Figure 3d shows the machined zone on the Al-SiC workpiece. The merits of powder mixed EDM have been witnessed from literature. Hence, the SiC (220 mesh) abrasive particles and graphite particulates (400 mesh) were used in 30 g/L for circulating through tool while machining. All the experimentation trials were conducted with magnetic field intensity, current, duration of pulse, volume percentage of SiC and type of dielectric medium (Table 2) as prominent parameters. The material removal rate, average surface roughness (Ra), and micro-hardness were evaluated as the response outcomes in this study. The surface roughness value was recorded with surface roughness tester (Surftest SJ-400, Mitutoyo America Corporation, Aurora, IL, USA) at three distinct positions of the machined surface and mean was considered.

To achieve accuracy in output responses, two replications were performed at random order to find out the mean. The Taguchi’s experimental design matrix [22] assisted the authors to identify and scrutinize the prominent parameters for this study. The orthogonal array of Taguchi’s experimental design reduces number of experimental trials to measure the effect of parameters included in the study. Valid conclusions during ED machining of Al-SiC MMCs were drawn and the factor assignment was done using Minitab-17 software. 

The selection of Taguchi’s orthogonal array depends on the number of factors (i.e., process parameters, herein 6 factors) and interactions of interest (herein, 1 interaction) and the number of levels of process parameters (i.e., 3 levels, refer Table 2). The total degree of freedom (fa) for each factor is the number of levels (la) minus one i.e., $f_{a\;}=l_{a}-1\;\left(i.e.,\;2\;\mathrm{for}\;\mathrm{each}\;\mathrm{factor}\right)$ and the degree of freedom for interaction is the product of interacting factors degrees of freedom i.e.$,\;f_{\mathrm{aXb}}=(f_{a})\left(f_{b}\right)\left(\mathrm{Herein}\;2*2\right)$. The minimum required degree of freedom in the experimental design is the sum of the entire factors and the interaction’s degree of freedom. Thus, the degree of freedom selected array (f_OA_) has must satisfy the inequality $f_{\mathrm{OA}}\geq \left(f_{\mathrm{aXb}}+f_{a}\right)$. Thus L27 orthogonal array was selected for the present study.

For the critical discussion on MRR, MH and SR of the powder mixed electrical discharge machining process; the results of the investigations are represented graphically. The various outcomes were analyzed through ANOVA to test the significance of model adopted. Prior to experimentation, the workpiece was designated as negative and tool with positive polarity. A precision electronic balance (Citizen, CY220, Mumbai, India) was used to measure workpiece weights before and after the machining. The investigation consisted of 27 distinct trials that helped to investigate the material removed, micro-hardness and surface quality of machined Al-SiC metal matrix composite using various input variables as tabulated in Table 3. This corresponding table constituted the input variables opted as a part of standard L27 orthogonal array control log and recorded values of responses for individual machined surface. The observed values, with and without magnetic field were recorded in designated columns as MRR, MH and SR output. 

## 3. Results and Discussion

### 3.1. Influence on the Material Removal Rate (MRR)

The experimental design matrix and results obtained, thereafter, are demonstrated in the Table 3. It is clear from Table 4 and Figure 4 that MRR is a function of current, duration of T-on/off and externally applied magnetic field. It has been exhibited that the magnetic field intensity was the most significant input parameter that affected the MRR steeply, followed by peak current and type of workpiece. Duration of pulse and dielectric medium, on the contrary, did not turn to be the significant parameters to influence MRR. At high discharge current, the enhanced MRR can be attributed to an improved volume of erosion in the maximum magnetic field intensity due to the high spark energy. The workpiece variant 1 (37 %vol. fraction of SiC) with comparatively lesser vol.% of SiC particulates resulted in enhanced MRR. This is attributed to the increased conductivity of workpiece owing to decreased vol.% of SiC particulates resulting in abrupt material removal. Furthermore, Figure 5 represented the surface morphology of work pieces of Al-37% SiC (Figure 5a) and Al-63% SiC (Figure 5b) after electrical discharge treatment. Figure 5a morphology witnesses the ablation as a mechanism of material removal of machined surface, whereas the Figure 5b indicates the melting and re-solidification of molten workpiece. The dense vol.% of SiC in the matrix exhibits shielding effect while machining and diminishes the MRR.

Moreover, the maximum material removal was witnessed in Trial 24, wherein 118% increase in MRR was observed in graphite powder mixed dielectric flushing in extreme magnetic field intensity compared to similar parametric conditions in Trial 5. The probable reason for increase in MRR could be the effective molten debris flushing on the machined surface due to the generation of Lorentz forces by the interplay of magnetic as well as electrical fields which enhanced the plasma pressure. Apart from this, injecting the powder mixed dielectric through the hole in the tube electrode enhanced its pressure, augmented the debris flushing. The interaction between different variables was not evident to be significant while powder mixed magnetic field assisted EDM.

### 3.2. Influence on the Micro-Hardness (MH)

The effect of various machining parameters and the surface conditions of the machined workpiece were analyzed by measurement of the indentation hardness. The micro-hardness measurements were carried out at the different locations of the machined surface as well as parent material, cross-sectionally. The machining parameters like magnetic field intensity, type of workpiece and pulse-on/off duration posed considerable effect on MH values, as shown in Table 5, except the discharge current and type of flushing which were the in-significant factors affecting the surface hardness of ED machined MMCs. It is clear from the graph (Figure 6) that machining of W/P-3 without magnetic field effect and in high-end pulse-on time resulted in the enhanced micro-hardness of the machined surface at minimum pause time. This is due to the reason that the higher concentration of SiC particulates resulted in oxides formation at high frequency (i.e., minimum pulse-off time) of sparks generated while machining.

The machined surface was further analyzed using X-ray diffraction method on PANalytical’s X’PertPro MPD (Netherlands). The X-ray spectra of the workpiece was characterized using Cu-Kα radiations (λ = 1.5406 A^0^) with the generator setting of 40mA and 45 kV. The X-ray diffractograms of the machined surfaces (Trial 8 and Trial 16) are shown in Figure 7. The peak pattern shows the formation of oxides such as Al_2_SiO_5_, SiO_2_, Si_3_W_5_ on the surface of machined MMCs, thus resulted in the enhanced hardness of machined surface [23]. On the contrary, the reduced surface hardness could be advantageous, wherein post-EDM additional processing such as grinding, grit blasting, etching etc., could be avoided [24]. Moreover, without magnetic field interference and in high current, more material deposition occurred that led to the enhancement in the micro-hardness (613.6%) as recorded (360.9 HV in Trial 8 as compared to 50.57 HV of parent workpiece) and 19.18% high as observed in Trial 23 (302.8 HV). There was no confinement of plasma and therefore more molten material deposited on the machined surface in the absence of magnetic field which was in line with previous studies related to magnetic field application. However, the interplay between various parameters and magnetic field were not significant enough affecting the output of the experimentation.

### 3.3. Influence on the Surface Roughness (SR)

The surface morphology of machined surface portrays a true signature of the tool-workpiece interaction while machining and material removal mechanism. The surface roughness was significantly influenced by magnetic field intensity and type of dielectric medium. The relations of SR with various process parameters are shown in Table 6 and Figure 8. The average surface roughness value for graphite powder mixed dielectric was witnessed as the lowest value. Alternatively, a rougher surface is achieved when machining is executed in 0.66 T magnetic field influence in SiC powder mixed dielectric medium. This can be related to the fact that at high magnetic field intensity abrupt material removal took place that turned the surface rough. Comparatively, a smoother machined surface is achieved when EDM is carried out in magnetic field intensity of 0.33 T. In this scenario, Lorentz forces developed while machining for little less duration comparatively with graphite kept the machining zone smoother and decreased the roughness. 

The interaction between various process parameters was statistically analyzed at 95% confidence level (i.e., *p* ≤ 0.05). It was observed that the interaction between magnetic field intensity (B) and T-on significantly affected the surface roughness. From Figure 9, it is depicted that the surface roughness enhanced at high magnetic field intensity and low T-on value. From the SEM analysis (Figure 10a), it can be observed that non-uniform surface with significant undulations is visible prominently during machining in SiC mixed dielectric medium in comparison to graphite mixed dielectric (Figure 10b). The high thermal conductivity of graphite particulates is responsible for distributing and dissipating uniform heat to the workpiece surfaces, thereby limiting the size of the craters thus produced. Owing to the externally applied magnetic field in powder mixed EDM process, the graphite powder particles under the effect of Lorentz forces crowded together to bridge the machining void in the dielectric fluid. Multiple plasma discharges are developed, as a result, from a unit pulse input avoiding deeper craters, resulting in improved surface quality [25]. Moreover, the generation of Lorentz forces with interference of external magnetic field into the EDM current density assisted in channelized path of current carriers from the tool to workpiece. 

## 4. Conclusions

The study was carried out on EDM of aluminum composite with plain and powder mixed flushing conditions. The following observations are enlisted.

The MRR is significantly affected by the machining parameters such as magnetic field environment, peak current and SiC% content of workpiece.The removal rate increased significantly with the incorporation of magnetic field intensity along with peak current.It is also evident that the decreased vol.% of SiC particulates led to a sharp rise in MRR. A 118% increase in MRR under the influence of magnetic field was observed in plain dielectric flushing when compared to identical parametric conditions in trials without magnetic field.An enhancement (613.6%) in the micro-hardness was witnessed due to the transfer of materials and formation of new phases while ED machining.The surface finish of machined MMCs was greatly affected by magnetic field intensity as well as type of dielectric. The surface finish improved steeply in graphite powder mixed dielectric flushing conditions at intermediate (0.33 T) magnetic field.

Future Scope: The literature studies depicted that aluminum matrix composites have been the major choice in the field of EDM of MMCs, but no significant work is available for the composite with other matrix phase such cobalt, steel etc. In addition to this, the experimental research can be extended and analyzed using various other levels of magnetic field of permanent magnets as well as rotating electromagnets.

## Figures and Tables

**Figure 1 micromachines-12-00469-f001:**
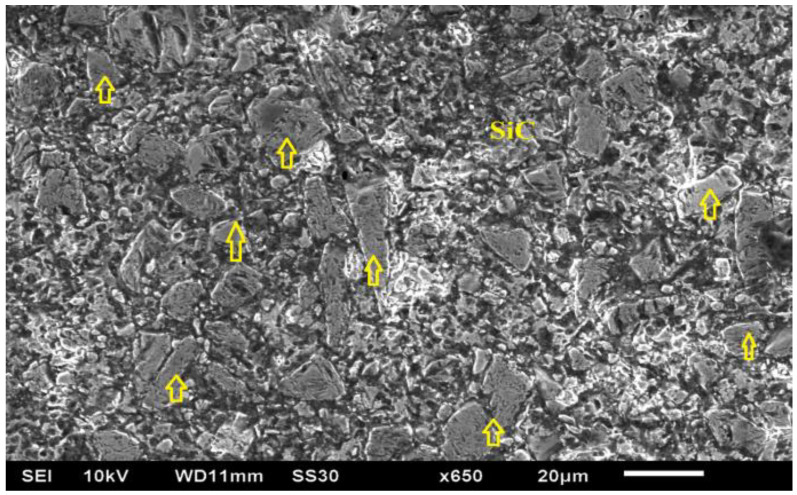
SEM (Make: JSM-6610LV Joel, Tokyo, Japan) of un-machined surface of Al-SiC workpiece at X650 (SiC particulates are highlighted with yellow color arrows).

**Figure 2 micromachines-12-00469-f002:**
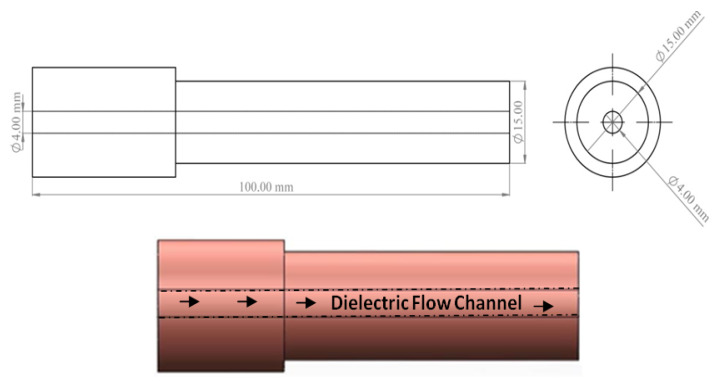
Copper electrode with inner hole for dielectric flushing.

**Figure 3 micromachines-12-00469-f003:**
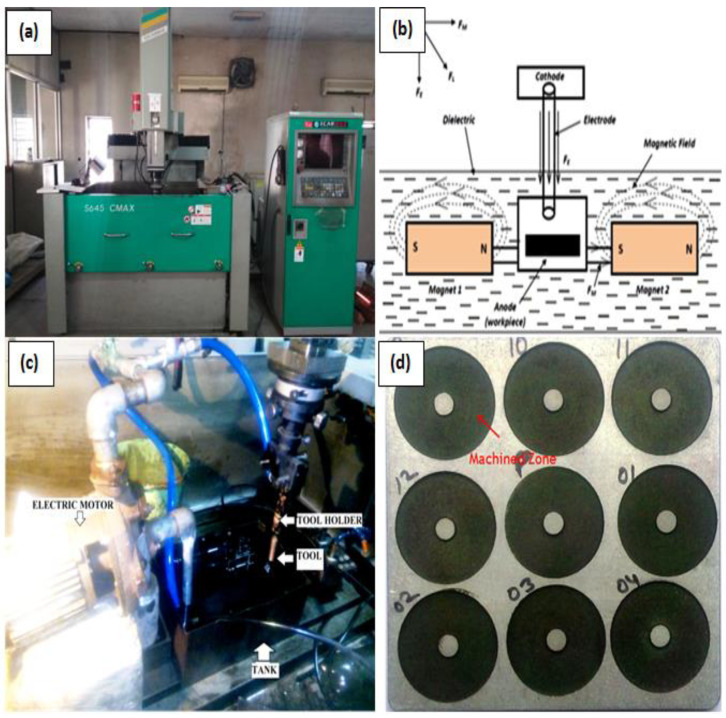
(**a**) EDM used for experimentation (**b**) Schematic diagram of MFAPEDM set-up and (**c**) Powder mixed dielectric set-up (**d**) machined Al-SiC workpiece.

**Figure 4 micromachines-12-00469-f004:**
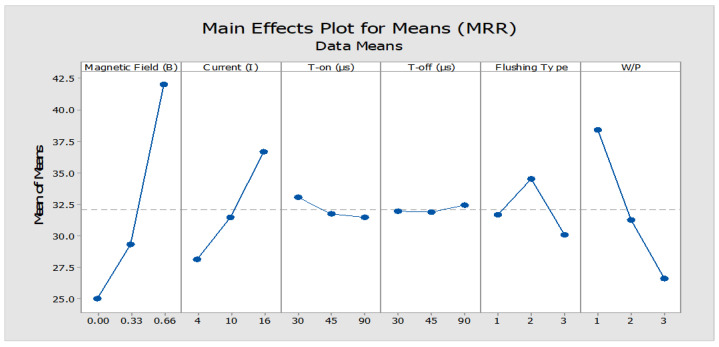
Main effect plot for material removal rate (mg/min).

**Figure 5 micromachines-12-00469-f005:**
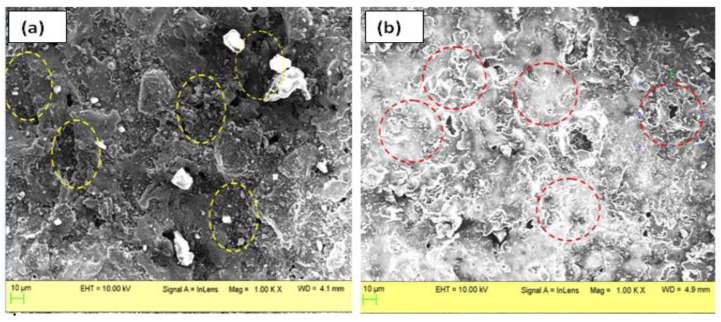
Scanning electron microscope (SEM) analysis of workpiece after machining (**a**) Al-37% SiC shows ablation mechanism of material removal; (**b**) Al-63% SiC shows melting as the dominating material removal mechanism.

**Figure 6 micromachines-12-00469-f006:**
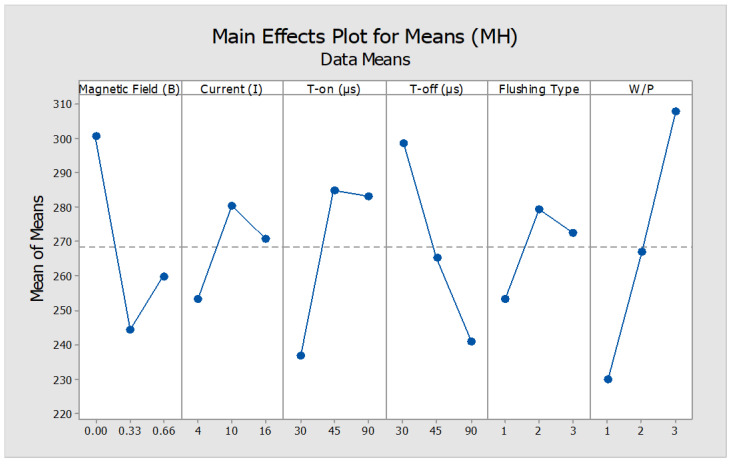
Main Effects plot for means for Microhardness (HV).

**Figure 7 micromachines-12-00469-f007:**
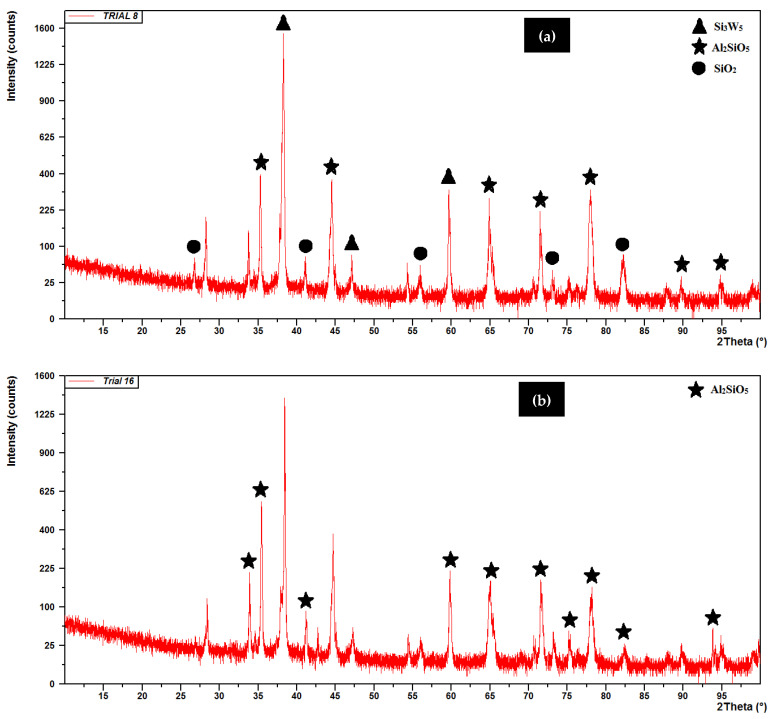
X-ray diffraction of (**a**) Trial 8 and (**b**) Trial 16.

**Figure 8 micromachines-12-00469-f008:**
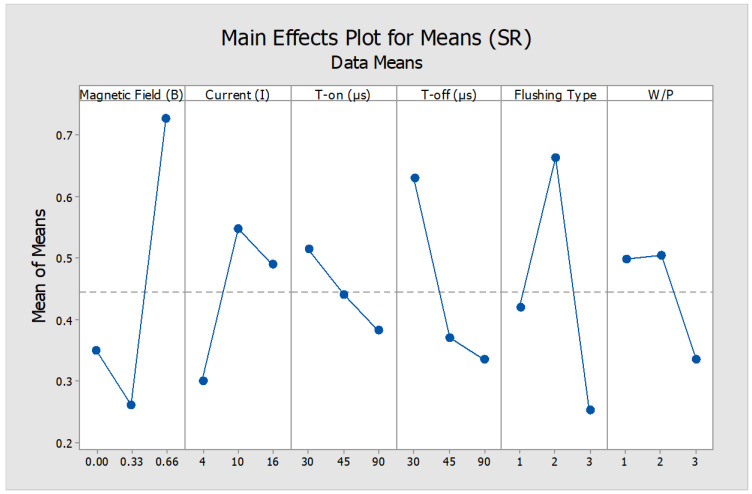
Main effects plot for surface roughness (µm).

**Figure 9 micromachines-12-00469-f009:**
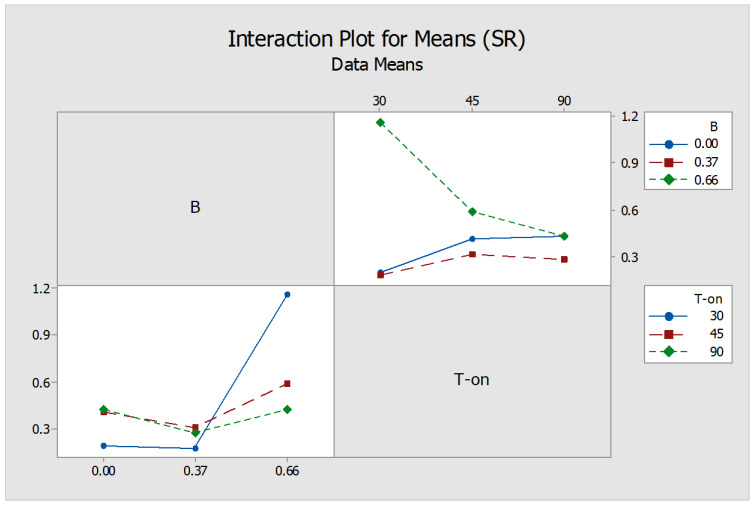
Interaction Plot between Magnetic field intensity (B) and T-on.

**Figure 10 micromachines-12-00469-f010:**
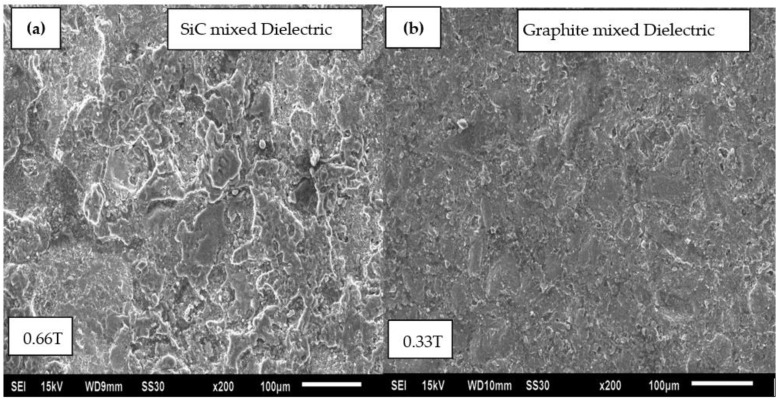
SEM analysis for comparative surface texture of various machined samples. (**a**) SiC mixed dielectric at 0.66T (**b**) Graphite mixed dielectric 0.33T.

**Table 1 micromachines-12-00469-t001:** MMCs properties (Source: CPS Tech, Norton, MA, USA).

Property	W/P-1	W/P-2	W/P-3
Aluminum Alloy (%)	63	45	37
SiC (%)	37	55	63
Thermal conductivity (W/mk)	170	190	190
Density (g/cc)	2.89	2.96	3.01
Specific Heat (J/gK) at 25 °C	0.808	0.786	0.741
Young’s Modulus (GPa)	167	167	188

**Table 2 micromachines-12-00469-t002:** Input parameters and their levels.

Variables/Notations	Level		
	1	2	3
Current (A)/I	4	10	16
Pulse-on (µs)/T-on	30	45	90
Pulse-off (µs)/T-off	30	45	90
Magnetic field intensity (T)/B	0	0.33	0.66
Dielectric medium	Plain dielectric	SiC mixed (220 mesh)	Graphite mixed (400 mesh)
Workpiece (W/P)	Al-37% SiC (W/P-1)	Al-55% SiC (W/P-2)	Al-63% SiC (W/P-3)

**Table 3 micromachines-12-00469-t003:** Experimental Factors and Results.

Process Parameters							Responses		
Trials	Magnetic Field (T)	Current (A)	T-on (µs)	T-off (µs)	Flushing Type	W/P	MRR (mg/min)	MH (HV)	SR (µm)
1	0	4	30	30	1	1	28.815	209.3	0.21
2	0	4	45	45	3	2	21.320	306.6	0.19
3	0	4	90	90	2	3	19.739	353.9	0.39
4	0	10	30	45	3	3	16.012	309.0	0.21
5	0	10	45	90	2	1	24.584	289.8	0.67
6	0	10	90	30	1	2	18.542	389.0	0.79
7	0	16	30	90	2	2	36.255	225.0	0.18
8	0	16	45	30	1	3	28.400	360.9	0.39
9	0	16	90	45	3	1	31.128	262.9	0.12
10	0.33	4	30	45	2	2	23.620	202.6	0.22
11	0.33	4	45	90	1	3	15.683	241.8	0.19
12	0.33	4	90	30	3	1	28.631	245.0	0.22
13	0.33	10	30	90	1	1	30.523	134.2	0.13
14	0.33	10	45	30	3	2	20.675	290.7	0.21
15	0.33	10	90	45	2	3	29.850	302.8	0.31
16	0.33	16	30	30	3	3	29.518	316.2	0.20
17	0.33	16	45	45	2	1	43.520	262.9	0.54
18	0.33	16	90	90	1	2	41.663	202.6	0.32
19	0.66	4	30	90	3	3	31.231	241.8	0.22
20	0.66	4	45	30	2	1	53.524	245.0	0.92
21	0.66	4	90	45	1	2	30.575	234.2	0.14
22	0.66	10	30	30	2	2	49.850	290.7	1.95
23	0.66	10	45	45	1	3	39.411	302.8	0.30
24	0.66	10	90	90	3	1	53.620	216.2	0.36
25	0.66	16	30	45	1	1	51.653	202.9	1.31
26	0.66	16	45	90	3	2	38.661	262.6	0.55
27	0.66	16	90	30	2	3	29.573	341.8	0.80

**Table 4 micromachines-12-00469-t004:** ANalysis of VAriance (ANOVA) for Material Removal Rate (MRR).

Scheme	DF	Seq SS	Adj SS	Adj MS	F-Value	*p*-Value
Magnetic Field (T)	2	1411.29	1411.29	705.646	12.95	0.001 **
Current (A)	2	336.98	336.98	168.488	3.09	0.077 *
T-on (µs)	2	12.71	12.71	6.357	0.12	0.891
T-off (µs)	2	1.61	1.61	0.806	0.01	0.985
Flushing Type	2	89.80	89.80	44.898	0.82	0.459
W/P	2	640.96	640.96	320.480	5.88	0.014 *
Residual Error	14	762.99	762.99	54.500		
Total	26	3256.35				

** Most significant, * Significant.

**Table 5 micromachines-12-00469-t005:** ANalysisof VAriance (ANOVA) for Micro-Hardness (MH).

Source	DF	Seq SS	Adj SS	Adj MS	F-Value	*p*-Value
Magnetic Field (T)	2	15,287	15,287	7643.6	10.91	0.001 *
Current (A)	2	3426	3426	1713.0	2.44	0.123
T-on (µs)	2	13,332	13,332	6665.9	9.51	0.002 *
T-off (µs)	2	15,191	15,191	7595.3	10.84	0.001 *
Flushing Type	2	3338	3338	1669.2	2.38	0.129
W/P	2	27,458	27,458	13,729.2	19.59	0.000 **
Residual Error	14	9811	9811	700.8		
Total	26	87,844				

** Most significant, * Significant.

**Table 6 micromachines-12-00469-t006:** ANalysisof VAriance (ANOVA) for Surface Roughness (SR).

Source	DF	Seq SS	Adj SS	Adj MS	F-Value	*p*-Value
Magnetic Field (T)	2	1.10890	1.10890	0.55445	8.30	0.008 *
Current (A)	2	0.30250	0.30250	0.15125	2.26	0.154
T-on (µs)	2	0.07783	0.07783	0.03891	0.58	0.576
T-off (µs)	2	0.47459	0.47459	0.23729	3.55	0.068 *
Flushing Type	2	0.76963	0.76963	0.38481	5.76	0.022 *
W/P	2	0.16805	0.16805	0.08403	1.26	0.326
B x T-on	4	0.92913	0.92913	0.23228	3.48	0.050 *
Residual Error	10	0.66784	0.66784	0.06678		
Total	26	4.49845				

* Significant.

## Data Availability

Not applicable.

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
