# Peer review of "Impact of Magnetic Field Environment on the EDM Performance of Al-SiC Metal Matrix Composite"

_micromachines, 2021, doi:10.3390/mi12050469_

Round 1

Reviewer 1 Report

Comment 1
Page 1 Abstract
aluminum-silicon carbide (Al-SiC) metal matrix composite. 
replace
Aluminum-Silicon Carbide (Al-SiC) metal matrix composite.

Comment 2
Page 2
and highlighted that a resultant 
force (FL) was witnessed. They demonstrated that the FL was due to the interplay of mag-
netic field (FM) and electric field (FE), which as a result increased the density of electrons 

replace

and highlighted that a Resultant 
Force (RF) was witnessed. They demonstrated that the RF was due to the interplay of Mag-
netic Field (MF) and Electric Field (EF), which as a result increased the density of electrons 

Comment 3
Page 3
First the authors mention Figures 2 and 3 in the text and then Table 2, 
while the order presented in the paper is first Table 2 and then Figures 2 and 3.
Move the Figures 2 and 3 before Table 2.  

Comment 4
Page 3
Figure 1
Insert arrows in the dimensions. Improve the quality of the Figure.

Comment 4
Page 3
Table 1
Density(g/cc) 
replace
Density (g/cc) 

Comment 5
Page 5
Table 3
Small number of experiments for six variables:
Magnetic Field (T)
Current (A)
T-on (µs)
T-off (µs)
Flushing Type
Workpiece (W/P)

Comment 6
Page 5
Insert a Figure with a typical workpiece before and after the EDM.

Comment 7
Page 6
Table 4. Analysis of variance (ANOVA) for MRR
replace
Table 4. ANalysis Of VAriance (ANOVA) for Material Removal Rate (MRR)

Comment 8
Page 6
Figure 6
The authors must explain the influence of T-on and T-off at the MRR. 
The figure shows that as the T-on increases (30 - 60 - 90 μs) the MRR decreases, 
while the opposite happens with the T-off. 
How is this influence possible?

Comment 9
Page 7
3.2. Influence on the MH
replace
3.2. Influence on the Material Hardness

Comment 10
Page 7
The authors did not cite the reference [20] in the text. 

Comment 11
Page 8
Table 5. Analysis of variance (ANOVA) for MH.
replace
Table 5. Analysis of variance (ANOVA) for Material Hardness.

Comment 12
Pages 8 - 9
Figure 6 title must be with the figure on the same page. 

Comment 13
Page 9
3.3. Influence on the SR
replace
3.3. Influence on the Surface Roughness

Comment 14
Table 6
Why did the authors use in this case the B x T-on?

Comment 15
The current experimental design and variable formulation implies that:
1) The effect when changing from flushing type 1 to flushing type 2 is equal to the effect of changing 
from flushing type 2 to flushing type 3 – similarly a change from flushing type 1 to flushing type 3 
should have double the effect compared to changing from 1 to 2 or 2 to 3. Is there any reason to support that?
2) For 6 explanatory factors and 3 responses, the standard factorial or random design methods would require 3^6=729 experiments 
whereas here only 27 are used. What are the consequences on the inference drawn? – For example, I failed to see a mean effect impact on e.g. 
MRR, however I did see that “Moreover, the maximum material removal was witnessed in Trial 24, 
wherein 118% increase in MRR was observed in graphite powder mixed dielectric flushing in extreme magnetic field intensity compared to similar parametric conditions in Trial 5.” 
which I think translates this impact by empirical means (not likely to happen again if the experiment is replicated?)
3) Refitting the models in Tables 4,5,6 only with the variables that came up significant,
does it produce a statistically valid model with acceptable p-values, normality of residuals and valid F tests?

Comment 16
References
Increase the number of the reference papers including (primarily) from MDPI Journals
The authors use 0 papers from Micromachines journal / 0 MDPI Journals / 23 papers from journals (References)
Τhe number for papers from MDPI journals
is considered insufficient (in reviewer's opinion).

Author Response

Authors sincerely thank the reviewer for the valuable comments and suggestions that helped to improve the quality of the paper. 

Reviewer 2 Report

Metal Matrix Composites (MMCs) are the lightweight materials possessing copious exceptional mechanical and physical properties for instance; high specific strength, stiffness and wear resistance. SiC particles reinforced aluminum matrix (Al-SiC) is one of the categories of such MMCs finding huge applications in the arena of aerospace and automotive and other various industries. The machining of difficult-to-cut materials such as ceramics, super alloys and composites has been a major challenge for manufacturing industries in today’s era. In this article, a hybrid magnetic field assisted powder mixed electrical discharge machining had been carried out on the aluminum-silicon carbide (Al-SiC) metal matrix composite. The aim of the study was to obtain higher surface finish, and enhanced material removal rate. The dielectric mediums employed were plain EDM oil, SiCp mixed and graphite powder mixed EDM oil for flushing through the tube electrode. I am pleased to send you moderate comments. The results and theme of this paper is quite interesting. The layout is clear and easy to understand. Generally, this manuscript makes fair impression and my recommendation is that it merits publication in this Journal, after the following major revision:

  1. The introduction should be reconstructed to present a coherent literature review. It may help the authors by answering the following questions: Why are these works relevant? Which specific problems were addressed? How are the previous results related with the latest work? What are the outstanding, unresolved, research issues? Answering the questions leads to the novelty of the proposed work naturally.
  2. Experimentation part, Although the results look “making sense”, the current form reads like a simple lab report. The authors should dig deeper in the results by presenting some in-depth discussion.
  3. In Table 3, the authors should give the explanations for the difference of data.
  4. An enhancement (613.6 %) in the micro-hardness was witnessed due to the transfer of materials and formation of new phases while ED machining. The surface finish of machined MMCs was greatly affected by magnetic field intensity as well as type of dielectric. The surface finish improved steeply in graphite powder mixed dielectric flushing conditions at intermediate (0.37 T) magnetic field. The authors should give some explanation on above conclusion.
  5. In this paper, it was observed that the machining variables for instance, discharge current, T-on/off duration and type of dielectric conditions remarkably affected the material removal rate, micro-hardness and surface roughness of the machined composite material. The recently published papers in the area of surface roughness of the composite material should be reviewed in Introduction section, (see [A fractal model for capillary flow through a single tortuous capillary with roughened surfaces in fibrous porous media, Fractals, 2021, 29(1):2150017; Fractals, 2019, 27(7): 1950116]). Authors should introduce some related knowledge to readers. I think this is essential to keep the interest of the reader.
  6. Please, expand the conclusions in relation to the specific goals and the future work.

Author Response

(The authors gave the same response as above.)

Round 2

Reviewer 1 Report

Comment 1
Page 1
Affiliation - Full alignment
Check if the authors names is united (need space)
such as
EvgenySergeevichShlykov
replace with (?)
Evgeny Sergeevich Shlykov (??)

Comment 2
Page 1
Keywords:electrical discharge
replace (insert a space)
Keywords: electrical discharge

Comment 3
References - Format
1.    Author 1, A.B.; Author 2, C.D. Title of the article. Abbreviated Journal Name Year, Volume, page range.
Change the References according to the journal's instructions 
Volume (Italics) and insert space.
Format References: 7, 10 and 11.

Comment 4 
Page 2
References should be numbered in order of appearance and indicated 
by a numeral or numerals in square brackets—e.g., [1] or [2,3], or [4–6]. 
After Reference [17], the authors mention [25].
The authors must renumber. Change reference [25] to [18] and renumber.

Comment 4 
Page 2
surface roughness (SR) [18]
replace
Surface Roughness (SR) [18]  

Comment 5 
Page 2
‘Lorentz force’ (LF) 
replace
‘Lorentz Force’ (LF) 

Comment 6 
Page 3
First the authors mention Table 1 in the text and then Figure 1, 
while the order presented in the paper is Figure 1 and then Table 1.
Move the Table 1 before Figure 1.  

Comment 6 
Page 3
The authors must insert mode details for SEM (model).

Comment 7 
The authors did not cite the reference [20] in the text. 
Response: Authors sincerely regret Typos. The reference [20] is now [21] and it is cited in the manuscript
The authors did not cite the reference [21] in the text. 

Comment 8
Insert a Figure with a typical workpiece before and after the EDM.
Response (Authors): The figure 1 is added in the manuscript.
Insert a Figure with a typical workpiece before and after the EDM (macroscopic point of view).

Comment 9
Table 6
Why did the authors use in this case the B x T-on?
Response: The authors investigated the interaction of various process parameters. The insignificant parameters are excluded in the study. Table 6 represents that B x T-on plays significant impact on SR (P=0.05*) with the confidence of 95%.
The authors response must be explained in the text.

Comment 10
The authors must explain in the text better the statistical analysis of the experiments. 

Author Response

(The authors gave the same response as above.)

Reviewer 2 Report

It is ok.

Author Response

(The authors gave the same response as above.)
